# Abundance and Growth of the European Eels (*Anguilla anguilla* Linnaeus, 1758) in Small Estuarine Habitats from the Eastern English Channel

**Jérémy Denis** [1,*] , **Kélig Mahé** [2] and **Rachid Amara** [1]

1 Université du Littoral Côte d'Opale, Université Lille, CNRS, IRD, UMR 8187, LOG, Laboratoire d'Océanologie et de Géosciences, F-62930 Wimereux, France
2 IFREMER, Fisheries Laboratory, 150 Quai Gambetta, BP 699, F-62321 Boulogne-sur-Mer, France
* Correspondence: jeremy.denis@univ-littoral.fr; Tel.: +33-321-996-427

**Abstract:** Abundance and growth of the European eel from six small northern French estuaries during their growth phase were examined to explore variations according to the local habitat characteristics. The length–weight relationships and growth models fitted to length-at-age back-calculated otolith growth increments were used to compare the growth. Higher abundances were observed in the smaller estuaries (2.4 to 10.5 ind. fyke nets 24 h$^{-1}$). The eel length ranged from 215–924 mm with an age range of 4–21 years. There was no significant difference in fish eel lengths or age except in the Liane estuary where the individuals were larger. The length–weight relationships showed an isometric or positive allometric growth in most estuaries. The Gompertz growth models, which best fits the growth, showed no significant differences between estuaries except for female eels from the Liane and the Somme estuaries where the growth performance index was higher. The estimated annual growth rate varied from 2.7 to 115.0 mm·yr$^{-1}$ for female and from 4.4 to 90.5 mm·yr$^{-1}$ for male. The present study shows that eels in the six estuaries had CPUE and growth rates similar to those previously reported in larger habitats. These results reinforce the idea that small estuaries are important habitats that contribute significantly to the eel population and, therefore, play an essential role in conservation strategies for European eel.

**Keywords:** length–frequency distribution; age; sex-ratio; hydro-morpho-sedimentary characteristics

## 1. Introduction

The European eel (*Anguilla anguilla* L. 1758) is an emblematic species, considered critically endangered by the IUCN and has a complex life cycle. During its continental life phase (i.e., growth phase), the European eel has a large geographic distribution from Northern Europe to North Africa [1]. The growth phase of eels is essential for transoceanic migration, fecundity and reproductive success [2–4]. Variations of growth depend of longer distances from the spawning site [5,6], latitudes and/or temperature range [7,8], habitat productivity as well as differences in fish density [9]. It has also been reported that eel growth depends on habitat use [7,10]. The European eel is a facultative catadromous species [11] that occupies a variety of freshwater, brackish and marine habitats during the growth phase [5,10,12]. Three main migratory tactics have been identified, with the marine and brackish resident, the freshwater resident, and the "migrant" eel that moves from one habitat to another [13–15]. Marine and brackish resident eels generally have a higher growth rate than freshwater residents [7,10,16], due probably to the high potential of available prey (mainly marine macrozoobenthos species) and the length of the growing season (i.e., a temperature above 12 °C corresponding to the growth limit temperature of eel [17]) compared to freshwater habitats.

Knowledge of the growth phase of eels is still limited in brackish and shallow coastal habitats [18], despite the fact that these habitats can be considered as important for the

European eel, especially in brackish systems such as estuaries, which host higher eel densities than freshwater habitats [19]. Studies of individual fish and population growth are important for understanding life-history strategies [20], and are necessary for analyzing eel population dynamics, stock assessment and management across the range of habitats used. The few studies on eel population growth have focused on differences between habitat use (i.e., marine, brackish and freshwater; [10,21–26]), temperature ranges and/or latitudes [7,27], sexual dimorphism and/or silvering stages [7,16,22,28,29], as well as on habitat quality (i.e., contaminant concentrations) [30]. Habitat characteristics, such as water temperature and/or the geographical location (i.e., latitude), appear to be the most influential external factors on eel growth. Eels living in higher temperature habitats, particularly in southern Europe, grow faster than northern individuals, where temperatures are lower [21]. However, growth differences have also been attributed in some cases to local habitat characteristics such as hydro-morpho-sedimentary features (e.g., intertidal seascape [31], river or estuary [32]) or food-web and productivity [33–35].

To our knowledge, the variation of European eel growth in estuary habitats according to the local habitat characteristics has never been examined. In addition, investigations on eels focused almost exclusively on large estuary (e.g., in the Severn Estuary [30] and in the Gironde Estuary [10,25]), neglecting the role of small estuaries, which are the most numerous. Conservation and implementation of management measures for eel populations require an understanding of the importance of small estuaries for eels, particularly in terms of their carrying capacity and growth potential [36,37]. In the present study, we analyzed the European eel abundance and growth attributes in six small estuaries along the French coast of the eastern English Channel and explored whether local habitat characteristics influence them. Specifically, we estimated and compared eel abundances, sex ratio, length structure, length–weight relationship and growth models based on length-at-age back-calculated otolith growth measurements. Finally, we assessed the influence of an estuary's hydro-morpho-sedimentary and anthropogenic characteristics on both female and male eel's growth.

## 2. Materials and Methods

### 2.1. Eels Sampling

The European eels were sampled in six estuaries (i.e., Slack, Wimereux, Liane, Canche, Authie and Somme) along the French coast of the eastern English Channel (Figure 1 a). The authorization to collect fish in the estuaries and access to the field site was issued by the Interregional Directorate for the Eastern English Channel-North Sea (dram-npe@equipement.gouv.fr; Decision n°196/2019). This study was conducted in accordance with European Commission recommendation 2010/63/EU, on revised guidelines for the accommodation and care of animals used for experimental and other scientific purposes. Eels were captured in 2019 and 2020 during four sampling periods each year (February–March, May–June, July–August and October–November). Eels were sampled using two fyke nets (16 m long with a mesh size of 15 mm at the beginning, 10 mm in the middle and 8 mm at the cod end) deployed in each estuary at three or four stations (separated by 0.5 to 8 km depending on the estuary size; Figure 1b) along the salinity gradients. The fyke nets were installed along the shoreline at low tide for a period of $2 \times 24$ h.

### 2.2. Sampling Areas Description

The six estuaries are located at very close latitudes (i.e., between $50°13'31.78''$ N and $50°48'18.79''$ N) and have similarities in terms of water temperature and salinity ranges [38] but have their own hydro-morpho-sedimentary characteristics. The Slack, Wimereux and Liane estuaries are the smallest estuaries. The Slack, Wimereux and Liane estuaries have a mean annual flows of 0.64, 1.04 and 4.35 $m^3 \cdot s^{-1}$ (water agency hydro. eaufrance.fr) and a surface area of 21.8, 22.1 and 38.2 $km^2$ (IGN-F maps), respectively. These estuaries have a narrow mouth width of about 0.1 km, sheltering them from tidal action, are and characterised by a dominant substrate of mud/sand for the Slack and Wimereux, and

mud for the Liane [38]. The water quality of these estuaries is considered to be in medium ecological status but in good chemical status (SDAGE 2016-2021). The Slack and Liane estuaries include the existence of dams that delimit the lower and upper parts, leading to higher exposure to freshwater inputs. The Canche, Authie and Somme estuaries have higher mean annual flows of 12.1, 7.3 and 35.2 $m^3 \cdot s^{-1}$ and a surface area of 5.3, 11.9 and 41.7 $km^2$, respectively. They are composed mainly of sand and gravel sediments [32]. These medium estuaries have mouth widths of 4.6, 2.9 and 5 km, respectively, and are more exposed to tidal action [32]. Their ecological and chemical status is similar to that of the smaller estuaries, with the exception of the Somme estuary, which has a poor chemical status. Human activities are much more important in the medium estuaries, notably due to the presence of agricultural activities, industry and dams in the upstream (Naïades database: naiades.eaufrance.fr).

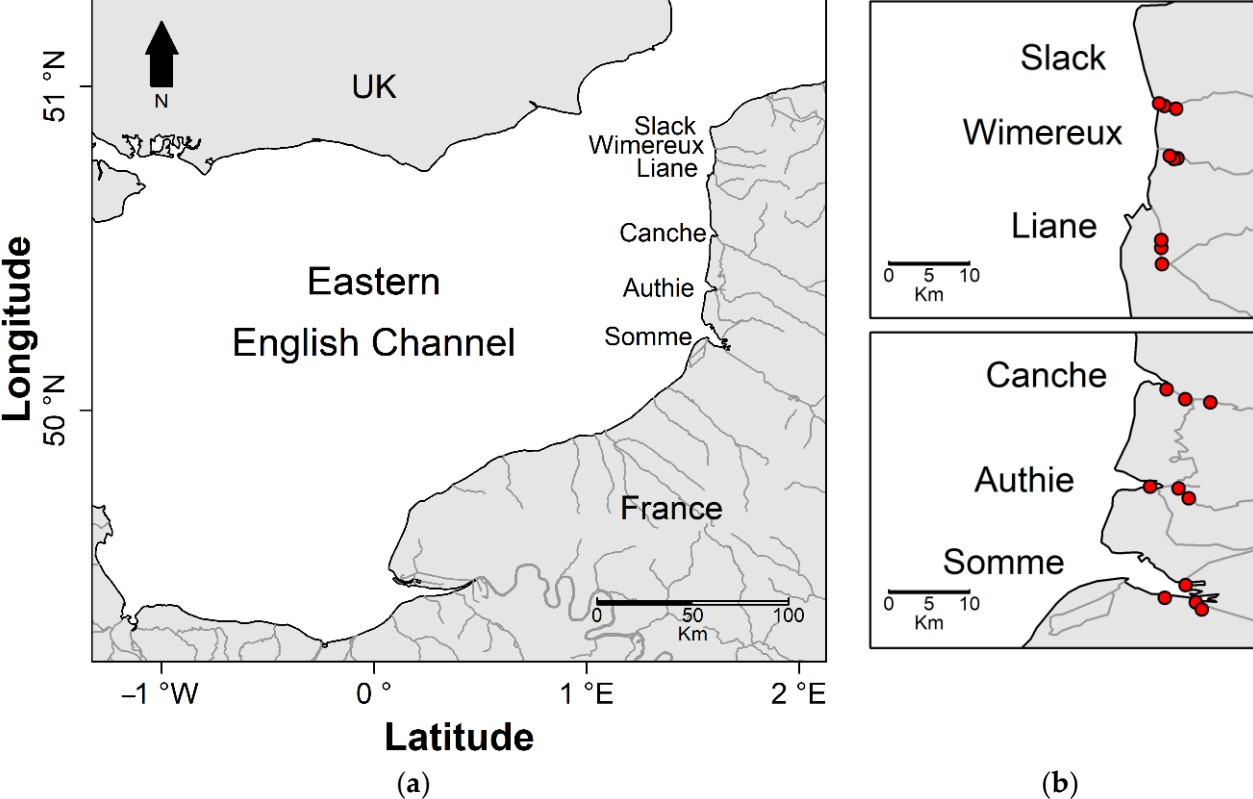

**Figure 1.** Location of (**a**) the six sampling estuaries along the French coast of the Eastern English Channel and (**b**) location of the three or four sampling stations per estuary (red dot).

*2.3. Sample Processing*

The eels captured were anaesthetized with eugenol solution (0.04 mL·L$^{-1}$; Thermo Scientific$^{TM}$) before being individually weighed (total Weight, W ± 1 g) and measured (Total Length, TL ± 0.1 cm). The sex of all eels caught was determined using the "silvering index" [39] based on measurements of body length, body weight, horizontal and vertical eye diameter (mm), and pectoral fin length (mm), and classified as undifferentiated growth phase (OH), female growth phase (FII), female pre-migrant phase (FIII), female migrating phases (FIV and FV) and male migrating phase (MII). A total of 226 eels (i.e., four to six eels for each sampling period, year and estuary) were then euthanized with a saturated eugenol solution and stored at −20 °C for otolith analysis. The other fish captured were released alive in the vicinity of the sampling station. Sex determination was confirmed for stored individuals by visual examination of morphological criteria [18]. Since female eels are larger than males at the same length due to sexual dimorphism [40], females and males

were analyzed separately. Undifferentiated individuals for which sex determination had not yet taken place were included in the growth in both female and male models.

The otolith radius ($R_o$), age (year) and growth of eels were determined on the right sagittal otoliths due to an asymmetry between the right and left otoliths [41] and to compare the results with other studies. The otoliths were extracted, cleaned with demineralized water, dried and embedded in Crystal bond® before polished with 200–800 µm micro-abrasive discs (LP Unalon®). The otoliths were examined under a stereomicroscope (oil-immersion, Olympus BX51) according to the international reading method [42]. Age, otolith radius and annual increments width (µm) were measured along the longest axis on the ventral side, from the elver mark corresponding to the beginning of the growth phase [42] to the edge. Annual increments in eel otoliths have been validated by considering a complete annual increment with a hyaline zone (i.e., cold period) and an opaque zone (i.e., warm period) [43,44].

### 2.4. Eel Abundance, Total Length and Weight

The abundance of eels was related to the fishing effort called Catch Per Unit Effort (CPUE), expressed in number of eels per gear and per unit of time (ind. fyke nets 24 h$^{-1}$). Since the data did not comply with the parametric assumption of normality (Shapiro–Wilk test) and homoscedasticity of variance (Levene's F test), the female and male eels' CPUE, total length and weight (combined and separately) were compared between estuaries each year (i.e., 2019 and 2020) using the non-parametric Kruskal–Wallis test and Dunn's multiple comparison test. The Kruskal–Wallis, Dunn test, Shapiro–Wilk test and Levene's F test were performed using the *stats* package in R 4.0.2.

### 2.5. Length–Weight Relationship

The eels' length–weight relationship (LWR) was fitted separately for female and male eels in each estuary. The length–weight relationship was calculated using the following Equation (1) and $\log_{10}$-transformed data were fitted using a least squared linear model following Equation (2):

$$W = aTL^b \tag{1}$$

$$\log_{10}(W) = \log_{10}(a) + b\log_{10}(TL) \tag{2}$$

where *W* is the total weight (g), *TL* is the total length (mm), *a* is the intercept or body shape coefficient and *b* is the slope and the growth coefficient (*b* value indicating an isometry growth when equal to 3.0; [45–47]).

### 2.6. Growth Model Selection

The linear relationship between total length and otolith radius was tested using a simple linear regression (i.e., $TL = a + b \times R_o$ [48]) for females and males separately.

Length-at-age (mm) was back-calculated using total length and otolith increment measurements and following the modified Fraser–Lee back-calculation procedure [49]:

$$TL_t = TL_c + (TL_c - TL_{bi}) \times \frac{(R_t - R_o)}{(R_o - R_{bi})} \tag{3}$$

where $TL_t$ is the length-at-age at age *t*, $TL_c$ is total length at capture, $TL_{bi}$ is length at the biological intercept, $R_t$ is otolith radius at age *t*, $R_o$ is otolith radius at capture, and $R_{bi}$ is otolith radius at the biological intercept.

The non-linear growth models were fitted to length-at-age data obtained from the back-calculated total length of female and male eels separately. Five most commonly used non-linear growth models were tested:

Von Bertalanffy model (vbp) [50]:

$$TL_t = TL_\infty \left(1 - e^{-k(t-t0)}\right) \tag{4}$$

Von Bertalanffy model forced at *t0* = 0 (vbt0p):

$$TL_t = TL_\infty - \left(TL_\infty . e^{-k.t}\right) \tag{5}$$

Von Bertalanffy model forced at $TL_{1i}$ (vbL1p):

$$TL_t = TL_\infty - (TL_\infty - TL_1) . e^{-k(t-1)} \tag{6}$$

Gompertz model (gp.p) [51]:

$$TL_t = TL_\infty . e^{\ln\left(\frac{TL_1}{TL_\infty}\right) . e^{-k(t-1)}} \tag{7}$$

Logistic model (log.p) [52]:

$$TL_t = \frac{TL_\infty}{1 + \left(\left(\frac{TL_\infty}{TL1}\right) - 1\right) . e^{-k.t}} \tag{8}$$

where $TL_\infty$ is the asymptotic length, $k$ is the rate at which the asymptote is reached and *t0* is the theoretical age (in years) at zero length. The value of *t0* has no biological significance [53]. The optimal growth model was selected based on the Akaike Information Criterion (AIC; [54,55]). The $TL_\infty$ and $k$ parameters of the selected growth model were used to characterize the female and male eels' growth [56,57] separately in each estuary.

In order to compare the selected growth model between estuaries, the growth performance index (mm·yr$^{-1}$, [58]) for female and males, was calculated separately:

$$\Phi = \log_{10}k + 2.31 \, log_{10}TL_\infty \tag{9}$$

where $\Phi$ is the growth performance, $k$ and $TL_\infty$ are the growth parameters of the selected model equation.

### 2.7. Growth Increments

Back-calculation of annual growth increments (mm·yr$^{-1}$) was calculated, separately for female and male, using the formula:

$$I_t = TL_t - TL_{t-1} = (TL_\infty - TL_{t-1})\left(1 - e^{-k}\right) \tag{10}$$

where *It* is the growth increments at age $t$, and $TL_{t-1}$ is the length-at-age $t - 1$. $k$ and $TL_\infty$ are the growth parameters of the selected model equation for female and male by estuary.

Normality and homoscedasticity of female and male eels' growth rates were assessed using a Shapiro–Wilk test ($p < 0.05$) and a Levene's F test ($p < 0.05$), respectively. Since the data did comply with the parametric assumption of normality and homoscedasticity of variance, two-way analysis of variance (ANOVA) and multiple Tukey (HSD Tukey) tests were used to compare growth rates between estuaries. To compare annual growth increments of eels of the same age between estuaries with a minimum of five individuals per estuary, annual growth increments from 1 to 10 years-old for females and from 1 to 7 years-old for males were considered in the tests. The ANOVA and HSD Tukey tests were performed using the *stats* package in R 4.0.2.

### 2.8. PCA Analysis

The variation of eel growth in estuary habitats according to the local habitat characteristics was analyzed using a Principal Component Analysis (PCA). The PCA was performed to determine how the spatial difference between female and male eels' growth parameters ($b$, TL$_\infty$, $k$, TL$_1$, $\Phi$, and $I_t$) could be explained by hydro-morpho-sedimentary and anthropogenic parameters (surface area, mean annual flow, wave exposure, mouth depth, substrate, mouth width, number of dams and chemical status) in the six estuaries.

The growth parameters were used as input variables and the hydro-morpho-sedimentary parameters were considered as quantitative supplementary variables. The PCA was used to summarize the variables into principal components and the relationship between variables was measured with the linear correlation coefficient (Pearson). The distance between observations (i.e., six estuaries) was based on the Euclidean metric. The growth, hydro-morpho-sedimentary and anthropogenic variables were log-transformed (log + 1) to reduce the skewness of the distribution, then centered and reduced before analyses. The PCA was performed using *FactoMineR* [59] package in R 4.0.2.

## 3. Results

### 3.1. Eel Population Characteristics

The eel development stages were mainly undifferentiated (50%), followed by females (40%) and males (10%) (Figure 2a). The sex-ratio was favorable for females in all estuaries and particularly in the Canche (23%), Authie (36%) and Somme (45%) estuaries. Males were more abundant in the Slack (23%), Wimereux (21%) and Liane (25%) estuaries. Eel lengths ranged from 215 to 924 mm and the majority of individuals (73%) were small, ranging from 250 to 500 mm (Figure 3), except for the Liane estuary, where most of the individuals were larger, with a majority of individuals (85%) between 300 and 650 mm. These differences were similar between the two years (Figure 3).

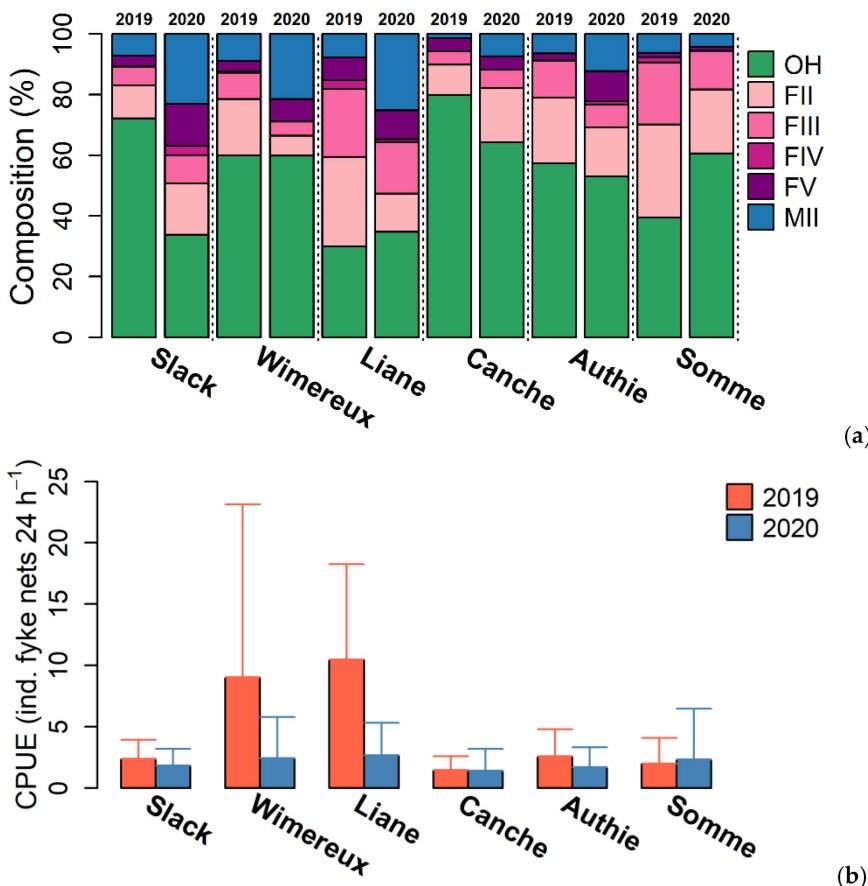

**Figure 2.** Variation of (**a**) the percentage of individuals per silvering stage (OH undifferentiated growth phase, FII female growth phase, FIII female pre-migration phase, FIV and FV female migration phases and MII male migration phase) and (**b**) means ± standard deviations in CPUE (ind. fyke nets 24 h$^{-1}$) of the eels in the six estuaries during 2019 and 2020.

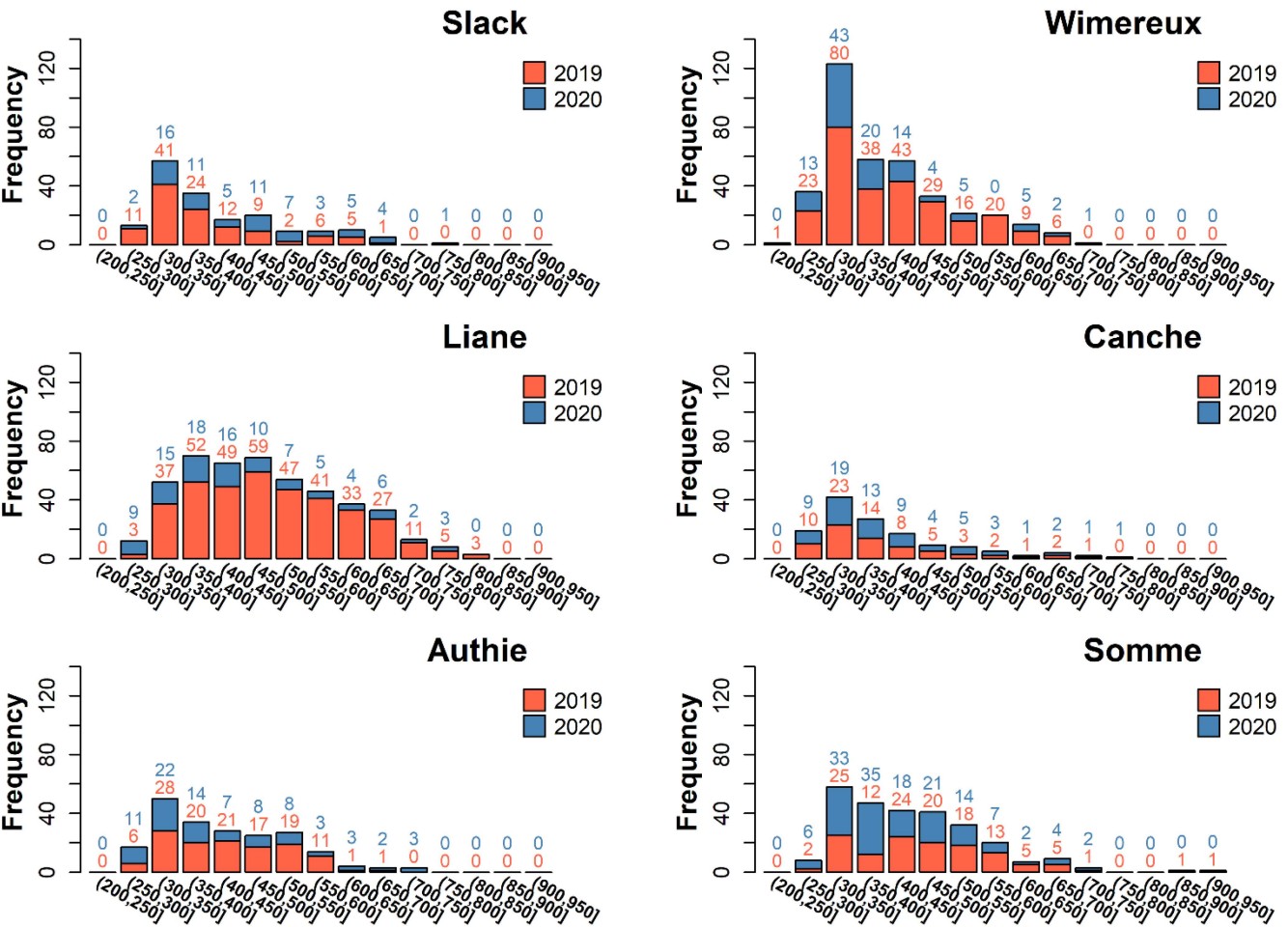

**Figure 3.** Total length–frequency distribution (TL in mm) of eels measured in the six estuaries during 2019 and 2020. The number of individuals measured is shown above each bar for 2019 (red) and 2020 (blue).

The CPUE means were significantly different across estuaries (Kruska–Wallis test, $p = 0.017$) with higher abundances in the Wimereux ($2.4 \pm 3.4$ to $9.0 \pm 14.1$ ind. fyke nets 24 $h^{-1}$) and Liane estuaries ($2.7 \pm 2.6$ to $10.5 \pm 7.6$ ind. fyke nets 24 $h^{-1}$) for both years (Figure 2b). Mean abundances were significantly different between estuaries for both females (Kruskal–Wallis test, $p = 0.0252$) and males (Kruskal–Wallis test, $p = 0.0335$). The abundances were significantly higher in 2019 than in 2020 (Kruskal–Wallis test, $p = 0.009$).

The total length of the eels analyzed ranged from 260 to 924 mm for females (mean: $463.6 \pm 135.8$ mm) and from 260 to 494 mm for males (mean: $361.8 \pm 51.4$ mm), with no significant differences in mean length between the six estuaries (Kruskal–Wallis test, $p = 0.05$ and 0.16, respectively; Table S1). The female and male eels' total weight also showed no significant differences between estuaries (Kruskal–Wallis test, $p = 0.07$ and 0.38, respectively). Values ranged from 32 to 987 g for females (mean: $238.2 \pm 237.1$ g) and from 32 to 247 g for males (mean: $88.3 \pm 41.1$ g) (Table S1). There was no difference in age between estuaries for either females (Kruskal–Wallis test, $p = 0.3583$) or males (Kruskal–Wallis test, $p = 0.9014$), with a mean age of $8.4 \pm 2.9$ years and $6.8 \pm 1.5$ years, respectively (Table S1).The length–weight relationship indicated that the initial body shape coefficient *a* varied from $-17.2$ to $-12.8$ for females and from $-13.5$ to $-11.7$ for males, while the growth coefficient *b* ranged from 2.9 to 3.6 and 2.7 to 3.0, respectively (Table 1). The eels from each estuary showed a significant relationship between total length and weight (Table 1). Almost all the analyzed female eels had a positive allometric growth with *b* higher than 3.0, except in the Slack estuary with a slight negative allometry (2.9),

indicating a tendency to be thinner than individuals in other estuaries. Male eels in the Wimereux, Liane and Canche estuaries exhibited an isometric growth with *b* values equal to 3.0, while eels in the Slack, Authie and Somme estuaries showed a negative allometry (*b* = 2.8), suggesting more elongated fish (Table 1).

**Table 1.** Length–weight (TL-W) and length–otolith radius (TL-$R_o$) relationship parameters for female and male eels collected in six estuaries. The values of coefficient *a*, the intercept or initial growth coefficient and *b*, the slope i.e., the growth coefficient and 95% confidence limits (%95 CL). $R^2$ is the correlation coefficient.

| Tested Parameters | Estuary | Female | | | Male | | |
|---|---|---|---|---|---|---|---|
| | | **a** | **b** | **$R^2$** | **a** | **b** | **$R^2$** |
| | | **%95 CL** | **%95 CL** | | **%95 CL** | **%95 CL** | |
| TL-W | Slack | −12.8 | 2.9 | 0.82 | −11.7 | 2.7 | 0.93 |
| | | −19.4–6.3 | 1.9–4.0 | | −19.4–6.3 | 1.9–4.0 | |
| | Wimereux | −17.2 | 3.6 | 0.89 | −12.9 | 3.0 | 0.91 |
| | | −21.8–12.5 | 2.9–4.4 | | −21.8–12.5 | 2.9–4.4 | |
| | Liane | −17.0 | 3.6 | 0.89 | −13.4 | 3.0 | 0.92 |
| | | −20.3–13.7 | 3.1–4.1 | | −20.3–13.7 | 3.1–4.1 | |
| | Canche | −16.7 | 3.5 | 0.91 | −13.5 | 3.0 | 0.87 |
| | | −20.4–13.0 | 3.0–4.1 | | −20.4–13.0 | 3.0–4.1 | |
| | Authie | −14.3 | 3.2 | 0.95 | −13.0 | 2.8 | 0.94 |
| | | −16.4–12.2 | 2.8–3.5 | | −16.4–12.2 | 2.8–3.5 | |
| | Somme | −15.3 | 3.3 | 0.94 | −11.9 | 2.8 | 0.87 |
| | | −17.6–12.9 | 2.9–3.7 | | −17.6–12.9 | 2.9–3.7 | |
| TL-$R_o$ | Slack | 15.0 | 0.27 | 0.76 | 132.0 | 0.17 | 0.73 |
| | | −92.0–121.9 | 0.21–0.34 | | 68.7–195.4 | 0.12–0.22 | |
| | Wimereux | 41.9 | 0.25 | 0.75 | 209.0 | 0.11 | 0.32 |
| | | −42.6–126.4 | 0.20–0.31 | | 128.6–289.3 | 0.05–0.17 | |
| | Liane | −45.6 | 0.33 | 0.67 | 205.5 | 0.12 | 0.23 |
| | | −180.5–89.4 | 0.25–0.41 | | 65.2–345.8 | 0.02–0.22 | |
| | Canche | 17.5 | 0.30 | 0.70 | 174.1 | 0.14 | 0.30 |
| | | −86.1–121.2 | 0.22–0.37 | | 54.6–293.5 | 0.04–0.24 | |
| | Authie | 27.0 | 0.28 | 0.64 | 47.5 | 0.24 | 0.69 |
| | | −78.2–132.1 | 0.21–0.35 | | −52.4–147.4 | 0.17–0.31 | |
| | Somme | 22.4 | 0.30 | 0.71 | 198.6 | 0.13 | 0.43 |
| | | −94.8–139.5 | 0.23–0.37 | | 88.6–308.5 | 0.06–0.21 | |

*3.2. Growth Model*

A significant linear relationship was found between the female and male eels' total length and otolith radius in each tested estuary (Table 1).

The Gompertz growth model was chosen for both female and male eels, as it was the model that best fitted the eels' total length-at-age data (model with lowest AIC, 18185.89 for female and 8357.17 for male; Table S2 and Figure S1). The asymptotic length (TL$_\infty$) values were higher for female in the Liane and Somme estuaries (751 and 1047 mm, respectively) (Table 2; Figure 4a) and for male in the Slack and Wimereux estuaries (500 and 516 mm, respectively) (Table 2; Figure 4b). The rate at which the asymptote was reached (*k*) showed higher values in the Canche and the Authie for females (0.20 and 0.22, respectively), whereas for the male eels, *k* was higher in the Liane and the Somme (0.29 and 0.30, respectively). The growth performance index $\Phi$ of female eels was greater in the Liane and the Somme (59 and 61 mm·yr$^{-1}$, respectively), and similar for other estuaries (58 mm·yr$^{-1}$). The $\Phi$ varied little between estuaries for the male eels (55 to 56 mm·yr$^{-1}$) (Table 2).

**Table 2.** Parameters of the Gompertz growth model and 95% confidence limits (%95 CL) for female and male eels collected in the six estuaries with $TL_\infty$ the asymptotic length, $TL_1$ the length in the first year, $k$ the rate at which the asymptote is reached, and $\Phi$ the growth performance index (mm·yr$^{-1}$).

| Estuary | Female | | | | Male | | | |
|---|---|---|---|---|---|---|---|---|
| | $TL_\infty$ %95 CL | $TL_1$% 95 CL | $k$ %95 CL | $\Phi$ | $TL_\infty$ %95 CL | $TL_1$ %95 CL | $k$ %95 CL | $\Phi$ |
| Slack | 641 | 109 | 0.19 | 57.6 | 500 | 107 | 0.23 | 55.9 |
| | 574–708 | 97–122 | 0.15–0.22 | | 436–564 | 99–114 | 0.18–0.27 | |
| Wimereux | 701 | 111 | 0.16 | 57.9 | 516 | 106 | 0.20 | 55.7 |
| | 601–800 | 99–123 | 0.13–0.20 | | 416–616 | 98–115 | 0.15–0.25 | |
| Liane | 751 | 112 | 0.19 | 59.2 | 442 | 110 | 0.29 | 55.7 |
| | 647–855 | 98–126 | 0.15–0.23 | | 367–518 | 96–125 | 0.20–0.38 | |
| Canche | 621 | 107 | 0.20 | 57.5 | 437 | 104 | 0.27 | 55.2 |
| | 576–667 | 96–118 | 0.17–0.22 | | 391–482 | 97–111 | 0.22–0.31 | |
| Authie | 596 | 109 | 0.22 | 57.5 | 437 | 103 | 0.27 | 55.3 |
| | 523–669 | 96–122 | 0.17–0.26 | | 387–488 | 90–115 | 0.21–0.33 | |
| Somme | 1047 | 127 | 0.12 | 60.6 | 433 | 102 | 0.30 | 55.7 |
| | 873–1221 | 114–141 | 0.10–0.14 | | 370–497 | 88–116 | 0.22–0.39 | |

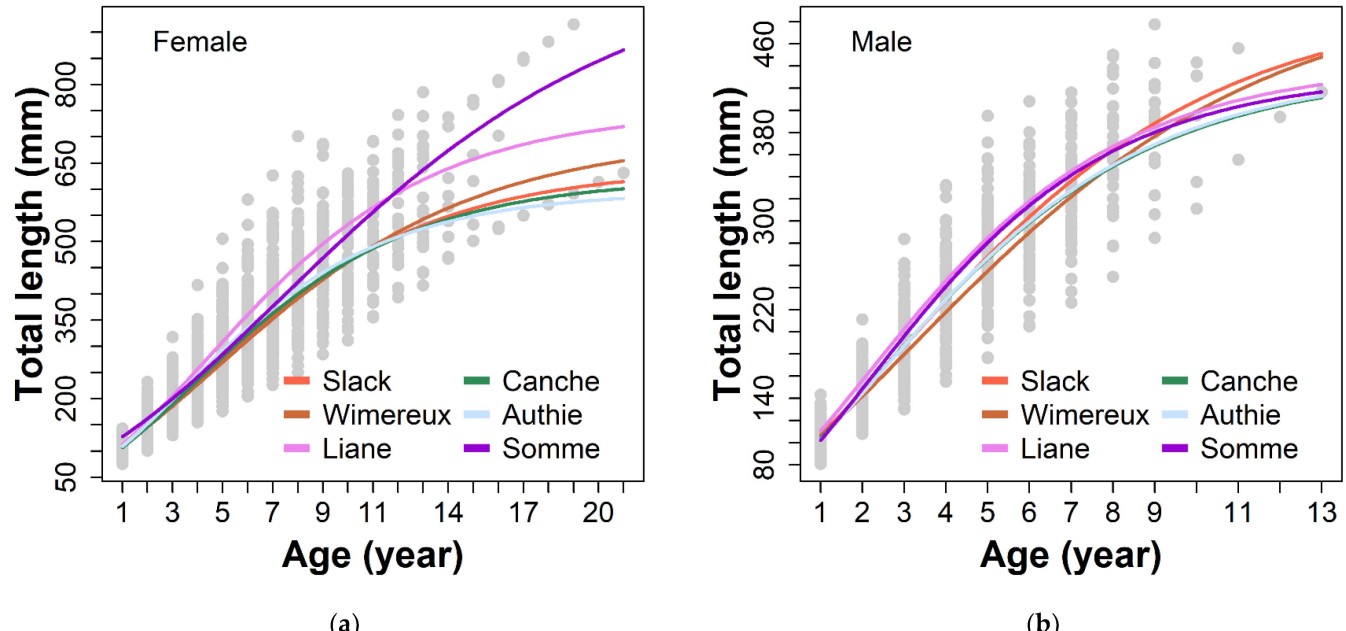

(**a**)                    (**b**)

**Figure 4.** Gompertz growth model (colored lines) fitted to eels' total length-at-age data collected in the six estuaries for (**a**) female and (**b**) male eels (grey dots).

*3.3. Annual Growth Rate*

The estimated annual growth rate varied from 2.7 to 115.0 mm·yr$^{-1}$ for females and from 4.4 to 90.5 mm·yr$^{-1}$ for males. Mean growth rate decreased rapidly with age; it was around 98.6 ± 8.1 mm·yr$^{-1}$ at one year-old and reached 38.5 ± 18.0 mm·yr$^{-1}$ at 10 years-old for the females (Figure 5a) and 79.8 ± 4.3 mm·yr$^{-1}$ at one year-old and reached 29.5 ± 10.3 mm·yr$^{-1}$ at 7 years-old for the males (Figure 5b). The ANOVA and HSD Tukey tests of female and male eels indicated that female and male eels' annual growth increments were significantly higher in the Liane and the Somme than in any other estuary (HSD Tukey test, $p < 0.001$; Table 3, Figure 5a). Female growth rates were significantly higher in the

Liane estuary (between $39.3 \pm 15.4$ and $110.9 \pm 2.6$ mm·yr$^{-1}$) and the Somme (between $64.2 \pm 10.9$ and $106.6 \pm 1.3$ mm·yr$^{-1}$) than in the other estuaries (between $22.5 \pm 16.4$ and $95.4 \pm 2.5$ mm·yr$^{-1}$).

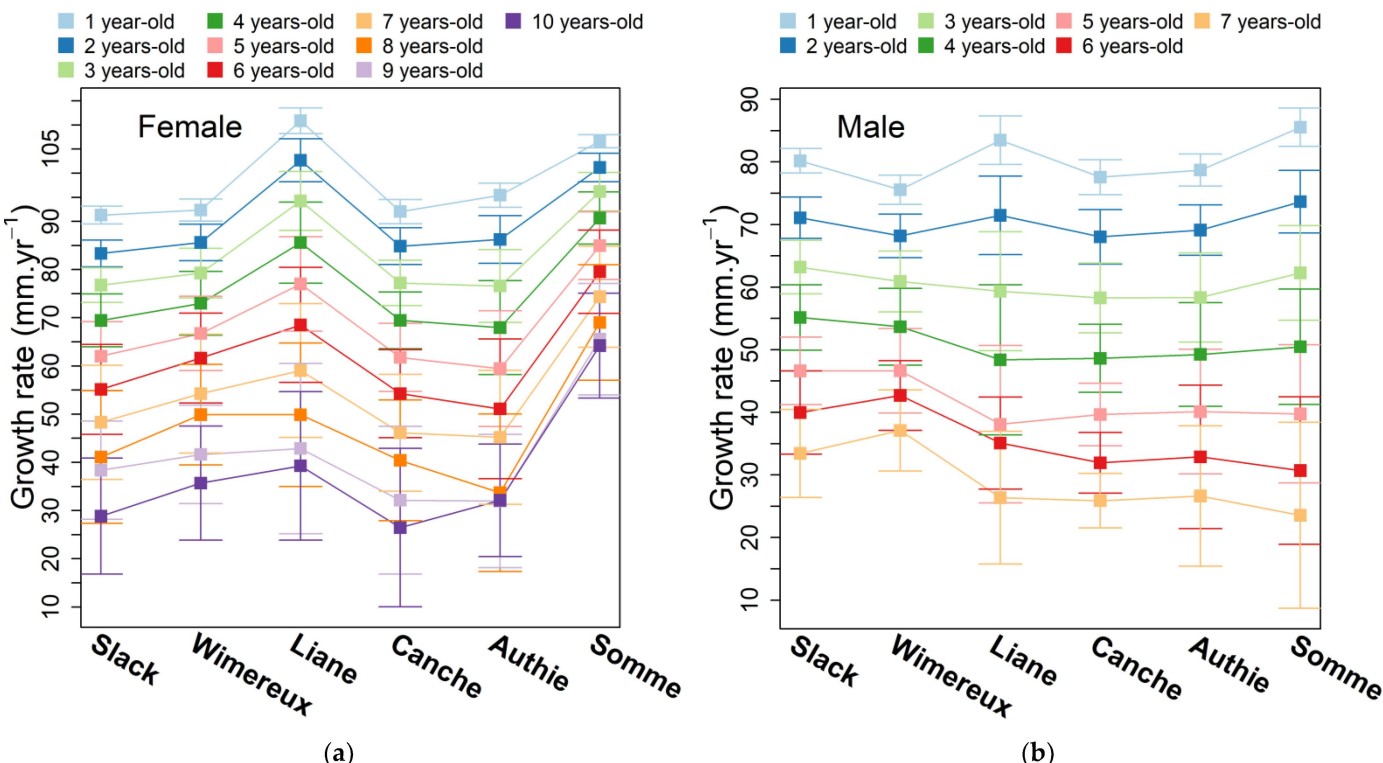

(**a**)                        (**b**)

**Figure 5.** Annual growth increments (mm·yr$^{-1}$) $\pm$ standard deviation from (**a**) 1 to 10 years-old female eels and (**b**) from 1 to 7 years-old male eels collected in the six estuaries.

**Table 3.** *p*-values of Tukey multiple comparisons tests on the estuary effects on annual growth increments of female and male eels collected in the six estuaries.

| Sex | Estuary | Slack | Wimereux | Liane | Canche | Authie |
|---|---|---|---|---|---|---|
| Female | Wimereux | <0.001 | | | | |
| | Liane | <0.001 | <0.001 | | | |
| | Canche | 0.999 | <0.001 | <0.001 | | |
| | Authie | 0.721 | <0.001 | <0.001 | 0.906 | |
| | Somme | <0.001 | <0.001 | <0.001 | <0.001 | <0.001 |
| Male | Wimereux | 0.706 | | | | |
| | Liane | <0.001 | 0.056 | | | |
| | Canche | <0.001 | <0.001 | 0.315 | | |
| | Authie | <0.001 | <0.001 | 0.746 | 0.974 | |
| | Somme | <0.05 | 0.458 | 0.947 | <0.05 | 0.195 |

Mean 1 to 7 years-old male eels' growth increments presented a similar pattern with higher values in the Liane and the Somme; however, their mean growth increments decreased more rapidly from 3 years-old and then reached the same values ($59.3 \pm 9.5$ and $62.3 \pm 7.6$ mm·yr$^{-1}$, respectively) as the other estuaries (between $58.4 \pm 5.6$ and $63.2 \pm 4.3$ mm·yr$^{-1}$, respectively) (Figure 5b and Table 3). Their mean growth increments became lower from 5 years-old ($38.1 \pm 12.6$ and $39.7 \pm 11.0$ mm·yr$^{-1}$, respectively) compared with those in the Slack, Wimereux and Authie estuaries ($46.6 \pm 5.4$, $46.6 \pm 6.7$ and $40.1 \pm 10.0$ mm·yr$^{-1}$, respectively).

### 3.4. Local Habitat Characteristics Influence on Eel Growth

The first two axes of the PCA explained 93.5% and 87.6% of the total female and male eels' growth variance, respectively (Figure 6). The PCA showed a clear separation between the variables $k$, $b$, and the variables $TL_\infty$, $TL_1$, $\Phi$ and $l_t$ for both female and male eels. All female eels' growth parameters, except for $b$, showed a positive correlation with the first axis (Figure 6a). $TL_\infty$, $TL_1$, $\Phi$ and $l_t$ from 1 to 10 years-old showed the highest positive correlations (between 0.81 and 1.00) and were associated with the estuary's chemical status (0.84), number of dams (0.86) and wave exposure (0.84). $k$ was negatively correlated with the first axis, whereas the $b$ (0.64) was positively correlated with the second axis and was associated with the mouth depth (0.84).

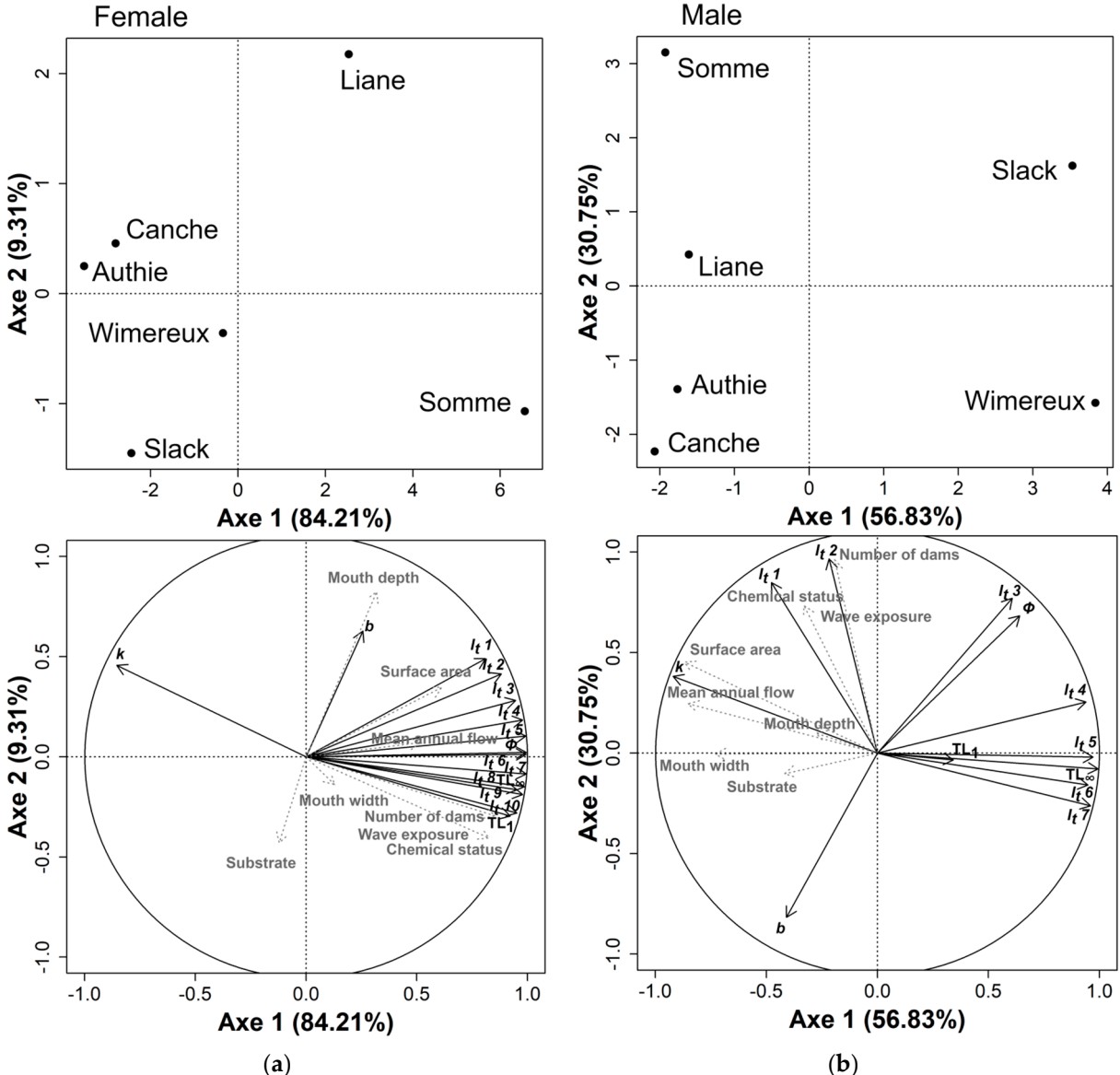

**Figure 6.** Observations (**top**) and correlation circle (**bottom**) of the first two axes of the Principal Component Analyses (PCA) on the growth of (**a**) female and (**b**) male eels ($b$, $TL_\infty$, $k$, $TL_1$, $\Phi$, and $l_t$; black arrows) in the six estuaries as a function of the hydro-morpho-sedimentary (surface area, mean river annual flow, wave exposure, mouth depth and width and substrate; grey arrows) and anthropogenic (number of dams and chemical status; grey arrows) variables.

Male eels' growth parameters (Figure 6b), $\Phi$ (0.64), $TL_\infty$ (0.99), and $l_t$ from 4 to 7 years-old (0.94–0.97), were positively correlated with the first axis, and while $k$ (−0.92)

was negatively correlated with the area surface ($-0.87$), mouth width ($-0.73$) and mean annual flow ($-0.85$). 1 to 3 years-old $l_t$ were positively correlated with the second axis ($0.77–0.97$) and were associated with chemical status ($0.73$), number of dams ($0.96$) and wave exposure ($0.73$), while $b$ was negatively correlated with the second axis ($-0.81$).

## 4. Discussion

The stock of the European eel is currently at its historical minimum. Effective conservation management of eels is hampered in particular by an incomplete understanding of the contribution of small estuaries. The present study highlights the importance of small estuaries located along the French coast of the eastern English Channel for European eel. Eels were most abundant in the smaller estuaries of the Wimereux and the Liane. CPUE values for the eels were in range with the highest densities reported in the ICES WGEEL database for larger rivers of Great Britain, France and Spain in the Interreg Atlantic and North Sea areas [60]. This suggests that small estuaries can support large eel populations. The sex ratio of eels caught in the six estuaries was in favor of females, with more than 80% of individuals compared to males. Generally, females are caught more upstream in freshwater, while males are significantly more important downstream in estuaries [61–63]. A sex ratio in favor of females in estuaries could be explained by the decrease in densities in recent years, which would have resulted in a higher proportion of females, since sex determination depends on density [18,64,65]. A possible change in the sex ratio may have occurred in recent years as a result of declining eel stocks [66,67]. The sex ratio of eels can be used as an indicator of population density, when environmental conditions are similar [68]. A higher proportion of males would indicate a higher population density at the time of sex determination [69,70]. However, sex determination may be affected by other parameters such as habitat quality and food availability [68]. In the six estuaries studied, we found a relatively higher proportion of males in the smaller estuaries of the Slack, Wimereux and Liane (23–53%) compared to the other estuaries (<25%). Sex ratios calculated from sex determination based on the silvering index [39], which allows a quick and non-invasive estimation of the sex of the eel, require more accurate verification by macroscopic observation of the gonads [71]. However, the results of sex ratios calculated with the same methodology between estuaries allow comparison of inter-estuary variations and highlights a significant contribution of small estuaries to eel populations. In terms of length structure, except for the Liane estuary which has larger eels, the eel lengths in other small estuaries do not vary compared to the other estuaries.

The length–weight relationship of eels showed that differences in body condition can be observed in both sexes, and that growth values are high in the six estuaries located along the French coast of the eastern English Channel. For the slope of the length–weight relationship, $b$ represents isometric growth with a value of 3.0 [45–47]. When $b$ is less than 3.0, growth will be negatively allometric and fish become thinner with increasing length, whereas when $b$ is greater than 3.0, growth will be positively allometric with larger fish reflecting optimal growth conditions. The $b$ measured in this study indicated that the growth of eels is optimal with values between 3.0-3.6 for females and 3.0 for males in most of the estuaries, except for the Slack estuary where values were below 3.0 for both males and females and for the Authie and Somme estuary only the males were below. These $b$ results are similar to those described for European eels in brackish and freshwater habitats further south in Europe [24,72,73], and higher than in high latitude [72,74]. These differences reflect the effect of latitude on eel growth, through a decrease with latitude with decreasing water temperature range and time of the growing season [8,18,22,75]. Other more local factors may also impact on the length–weight relations, such as density, food availability or pollution [76,77]. Nevertheless, these results suggest that the six northern French estuaries can be considered as offering optimal conditions for eel growth, close to that found in warmer and larger systems.

The growth model parameters are used to characterize growth [56,57]. The values of the parameter $k$ and $TL_\infty$ were highly variable between estuaries (0.12–0.30 and

433–1047 mm, respectively), and similar to other habitats in both brackish and freshwater conditions [21]. These variations in growth parameters are highly dependent on environmental conditions and demographic factors such as the range of sizes of individuals used to fit the growth models [78,79]. The growth performance index ($\Phi$) based on growth model parameters allows for better comparisons of growth between populations and/or habitats [57,79,80]. The growth performance index $\Phi$ of eels showed higher values between 57.5 to 60.6 mm·yr$^{-1}$ for the females and 55.2 to 55.9 mm·yr$^{-1}$ for the males, compared to other estuarine habitats with values between 46.4 to 51.4 mm·yr$^{-1}$ (i.e., in the Severn [30] and Guadalquivir estuaries [22]; the estuaries along the Southern Baltic [26] and the German Baltic coast [81]). In comparison with other brackish habitats such as Mediterranean lagoons, eels have growth values between 50.2 to 53.3 mm·yr$^{-1}$ for the females, 47.9 to 53.6 mm·yr$^{-1}$ for the males and 48.4 to 51.5 mm·yr$^{-1}$ for both (i.e., the Valli di Comacchio [29], Vaccarés-Impérieux, Fumemorte [24], Aveiro [82] and Valle Nuova lagoons [29]). Only eels from the Santo André lagoon in Portugal have higher growth, with 62.1 mm·yr$^{-1}$ for females and 59.5 mm·yr$^{-1}$ for males [21]. This highlights that the eels living in the six northern France small estuaries studied have much better growth performance than those living in brackish habitats in warmer latitude (i.e., further south in Europe), suggesting that parameters other than temperature may influence eel growth. Eels from freshwater habitats also presented lower growth values compared to our studied estuaries, with values between 46.6 to 48.5 mm·yr$^{-1}$ for the females and 31.6 to 46.1 mm·yr$^{-1}$ for the males (i.e., the Severn [9], Shannon [83], Frome [84], Koge Lellinge [85], Barrow [78] and Imsa rivers [86] and the Fertö lake [87]). This confirm that brackish habitats such as estuaries support higher growth rates than in freshwater habitats and offer probably more favorable conditions to support eel growth [10,24–26]. Estuarine habitats in temperate ecosystems are much more productive than rivers and allow for a longer period of higher temperatures and longer time of year, which underlines the importance of estuarine habitats for the growth of the eels [10].

The length–weight relations, growth models and growth rates were significantly different between the six estuaries located along the French coast of the eastern English Channel. In general, eels in the smaller estuaries (i.e., the Slack, Wimereux and Liane estuaries) exhibited rather high growth compared to the larger estuaries, except for the Somme that showed an exception for female eels with high growth. The water temperature range (or latitude) and distance to spawning sites are the two major factors that most explain differences in eel growth rate [7,8]. However, at a local scale, these factors do not vary significantly between the six local estuaries that were studied. Other factors may explain these growth variations, such as fish density and/or habitat productivity [9]. In terms of eel density, the Wimereux and Liane estuaries showed higher eel densities (9 $\pm$ 14 and 11 $\pm$ 8 ind. fyke nets 24 h$^{-1}$, respectively) compared to the other estuaries (between 1.4 $\pm$ 1.1 to 2.6 $\pm$ 2.2 ind. fyke nets 24 h$^{-1}$), which could lead to strong competition for food and thus limit optimal growth. Our results do not support such a hypothesis since growth rates were relatively higher in these smaller estuaries. Food limitation is one of the major factors influencing the growth of fish, especially juvenile fish in coastal nurseries [88]. Eels have opportunistic feeding plasticity with a feeding preference for macro-crustaceans, capable of feeding on other prey such as fish, mainly when benthic invertebrates are less available [63–65]. This dietary plasticity of eels leads to spatial differences in dietary, in particular in the six estuaries [89], with a diet more focused on fish in smaller estuaries and on macro-crustaceans in larger estuaries (unpublished data). These differences suggest that feeding on more energetic prey, such as fish, would help maintain optimal growth. A previous study also showed that the trophic ecology of eels in these estuaries may be influenced by the availability of macrozoobenthos species, which in turn depends on the hydro-morpho-sedimentary characteristics of the estuary [89]. On a local scale, the characteristics of the estuary may be a likely mechanism for indirect regulation of eel growth through food availability.

Several causes have been suggested to explain the decline in eel stocks, including habitat loss and dams that reduce accessibility to growth habitats, pathogens and pollution [76,90–93]. The presence of barriers to migration and alteration of water quality and their habitats have been suggested as the main causes which have led to their decline [60,94–96]. The six estuaries studied in the present study are less affected by anthropogenic pressures and considered to be of good chemical status and medium ecological status, except for the Somme estuary, which is rated as poor. In addition, only the Liane and Somme estuaries have dams that likely impact on the free movement of eel populations. No negative correlation was found between female eel's growth parameters and the hydro-morpho-sedimentary and anthropogenic variables analyzed. However, the growth of male eels seems to be impacted by some parameters such as the river flow, the surface area and the mouth depth, especially for the older ones (i.e., 3 to 7 years old). The rate ($k$) at which male eel's approach the population asymptotic size ($TL_\infty$) is positively correlated with these later parameters. Male eels being smaller than females may be more sensitive to habitat quality and to obstacles limiting accessibility to growth habitats. It has been shown that the alteration of the banks is likely to hinder the lateral ecological continuity between the main watercourse and the submerged areas (e.g., silt, intertidal flats, etc.) [97,98]. However, the small sample size of male eel's (representing only 10% of the catches) does not allow us to conclude anything regarding the effect of the hydro-morpho-sedimentary and anthropogenic variables analyzed on the growth of the males. Other anthropogenic factors, such as lateral habitat loss due to channelization and reinforcement of the riverbanks need to be to be considered in future studies. Nevertheless, the relative high growth performance measured in the six estuaries compared to other brackish water and estuarine habitats [21,22,24,26,30,81,99] supports the idea that the environmental conditions of these estuaries are favorable to eel growth.

## 5. Conclusions

CPUE and estimated growth of European eels in the estuarine habitats have highlighted the importance of small estuaries as a crucial habitat for supporting higher growth during their growth phase. The present study shows that eels in the six estuaries located along the French coast of the eastern English Channel had growth rates similar to those measured in brackish habitats further south and higher than those measured in freshwater habitats. In addition, CPUE values were in range with the highest densities previously reported in larger habitats. These results reinforce the idea that small estuaries are important habitats that contribute significantly to the eel population and, therefore, must be considered in both population status assessments, conservation and management strategies for the European eel [36].

**Supplementary Materials:** The following supporting information can be downloaded at: https://www.mdpi.com/article/10.3390/fishes7050213/s1, Figure S1: Non-linear growth models of female and male eels collected in the six estuaries; Table S1: Number of individuals (N) analyzed of female and male eels for the six estuaries from the North to the South and their mean ± standard deviation (sd) and range (min–max) of total length (mm), total weight (g) and age-at-capture (year); Table S2: Parameters of non-linear growth models for female and male eels collected in the six estuaries with TL∞: the asymptotic length, k: the rate at which the asymptote is reached, TL1: the length in the first year, t0 no biological significance and AIC the Akaike Information Criterion to select the optimal growth model.

**Author Contributions:** Conceptualization, J.D. and R.A.; Methodology, J.D. and K.M.; Validation, J.D., K.M. and R.A.; Formal analysis, J.D., K.M. and R.A.; Investigation, J.D.; Writing—original draft preparation, J.D., K.M. and R.A.; Supervision, R.A.; Funding acquisition, J.D. and R.A. All authors have read and agreed to the published version of the manuscript.

**Funding:** This research was funded by "Parc Naturel Marins des Estuaires Picards et de la Mer d'Opale" (DECISION N°2018-28 9 March 2018), and "European Maritime Fisheries Fund" and "Région Hauts de France" (PFEA621220CR0310022). This work also has been partially financially

supported by the European Union (ERDF), the French Government, the Région Hauts-de-France and IFREMER, in the framework of the project CPER MARCO 2015–2020 (https://marco.univ-littoral.fr/). The funders had no role in study design, data collection and analysis, decision to publish, or preparation of the manuscript.

**Institutional Review Board Statement:** The permission to collect fish in the estuaries and field site access was issued by the "Préfète de la region Normandie, préfète de la Seine Maritime, Direction interrégionale de la mer Manche Est-mer du Nord, Service Régulation des Activités et des Emplois Maritimes, Unité Réglementation des Ressources Marines (dram-npe@equipement.gouv.fr): Decision n°196/2019". This study was conducted in accordance with European Commission recommendation 2007/526/EC, on revised guidelines for the accommodation and care of animals used for experimental and other scientific purposes.

**Data Availability Statement:** The data presented in this study are available on request from the corresponding author.

**Acknowledgments:** The authors would like to thank K. Rabhi, M. Diop, M. Kazour and K. Boutin for their participation in the collection of the samples and A. Lheriau, R. Elleboode and A. Dussuel for their participation in the preparation of the otoliths. Finally, the authors are very grateful to the three anonymous referees and editor, who helped to greatly improve the initial manuscript with their positive and constructive comments.

**Conflicts of Interest:** The authors declare no conflict of interest.

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
