# Peer review of "Abundance and Growth of the European Eels (Anguilla anguilla Linnaeus, 1758) in Small Estuarine Habitats from the Eastern English Channel"

_fishes, doi:10.3390/fishes7050213_

Round 1
Reviewer 1 Report
Population structure and growth of the European eels (Anguilla anguilla L.) in estuarine habitats by Denis et al.
The study deals with sex, age, growth and length- growth relationships of the European eel from six northern French estuaries during their growth phase. I personally appreciate the paper but some improvements remain useful in abstract, introduction, Materials and Methods, results and discussion. The text still has some inaccuracies making it difficult to understand, their improvements would make the paper easier to read. In my opinion the manuscript still needs some improvements before any considerations in Fishes. Suggestions/comments are listed below as:
L34 the European eels has = the European eel has (as in L32 and L39.
L45-46 "...and the length of the growing season (i.e. high temperature range)..." this lacks precision and is hard to understand.
L45 "...high potential of available prey (mainly marine Macrozoobenthos)..." = "...high potential of available prey (mainly marine macrozoobenthos species)..." see also L391.
L69-70 "The objective of our study was to determine the population structure and growth of European eels..." It's confused, please improve for clarty.
L16-17 « A higher abundances and male proportions were observed in the smaller estuaries ». This contradicts and is not consistent with reported results L211-213 and also with discussion L303-304.
L17 Size structures ? This term seems imprecise and confusing. Please use a more precise and appropriate term.
L65-66 " ... according the local habitat characteristics ..." = "...according to the local habitat characteristics ...". See also L14, L71.
L68, 72 What meant by « small systems » ? See also L L357 « warmer systems ». It’s unclear. Please improve
L75 "The European eel were sampled ..." = "The European eels were sampled ..."
L76-77 "Each of these estuaries has similarities in temperature and salinity ranges ...". Which data ranges, nothing supports such assertions.
L97-98 Perhaps, it’s "The eels were captured during 2019 and 2020 during four sampling periods each year ..." = "Eels were caught in 2019 and 2020 during four sampling periods each year ..."
L109 Perhaps, « expressed in individuals per gear and per unit of time » = « expressed in number of eels per gear and per unit of time ».
L115-117 Why was the sex of euthanized eels not determined?
L209-210 Please express the Catch Per Unit Effort (CPUE) in number of individual eels per gear and per unit of time labeled as ind. fyke nets 24h-1.(see L109-110.)
L215 Perhaps, for clarty, « Eel sizes » = « Eel lengths »
L317-319 « However, in terms of population size structure, except for the Liane estuary which has larger eels, the size structure in other small estuaries do not vary compared to larger estuaries ». Please
L110-115 Authors stated that « The sex was determined using the “silvering index” [37] based on measurements of body length, body weight, horizontal and vertical eye diameter (mm) and pectoral fin length (mm), and classified as indeterminate growth phase (OH), female growth phase (FII), female pre-migrant phase (FIII), female migrating phases (FIV and FV) and male migrating phase (MII)». I'm not sure if this sexing methodology is the most appropriate due to the presence of a body size uncertainty zone between the two sexes and high proportion of indeterminate individuals. This is likely to change the reported sex ratio and growth chart results of eel stock in estuaries.
L118-119 What is meant by « The females and males were analyzed separately due to sexual dimorphism, since the female eels being larger than males at the same length » ? This is very confusing, please rephrase.
L119-220 « Undetermined individuals were analyzed with both females and males, as sex determination has not yet taken place ». The problem is that the eels have not been sexed. The methodology used is only reliable for large female eels. Moreover, indeterminate eels represent a significant proportion of the eels captured but they are less described in the results and even less discussed in the discussion.
L212-214 Authors mentioned that « The eels were mainly indeterminate (50%), followed by females (40%) and males (10%) (Figure S1). The sex-ratio was favorable for females in all estuaries and particularly in the Canche (18 to 28%), the Authie (35 to 36%) and the Somme (35 to 54%) estuaries ». This is not surprising because of the weakness of the methodology used to determine sex. I hope this is well discussed in the discussion section.
L237-239 « The age effect was significant, probably due to sexual dimorphism with females being larger than males at the same length, ... » . Please rephrase this sentence for the same reason mentioned above..
L303-316 Interpretation of the population sex ratio results requires caution, especially since half of the eels were sexually indeterminate. The variability in body size and age observed in eels suggests the existence of several cohorts in this population and therefore a certain limitation in the use of the sexing methodology. I think here should also be made the criticism of this methodology.
L418-420 . "These results reinforce the idea that small systems should be considered as essential habitats that also contribute significantly to the eel population." It’s unclear. Please rephrase.
Reviewer 2 Report
Dear Authors,
the manuscript ID fishes-1850112 is very interesting and present important data regarding populations structure and growth of the European eels in estuarine habitats from the Eastern English channel. As the paper presents data from this area, it should be stated in the title, then please add the location in the title of the manuscript .
The manuscript contains a lot of data and valuable concepts and discussion in defining differences among populations from the different estuaries considered. Despite this, it is very difficult to follow due to the scarce quality of the English language and style. For this reason it is my opinion that a native speaker english revision is mandatory to improve the readability of the manuscript. Moreover, it is suggested to use R-based shape analyses for assess differences among eel populations, sea for example "D’Iglio, C.; Albano, M.; Famulari, S.; Savoca, S.; Panarello, G.; Di Paola, D.; Perdichizzi, A.; Rinelli, P.; Lanteri, G.; SpanoÌ€, N.; et al. Intra- and interspecific variability among congeneric Pagellus otoliths. Sci. Rep. 2021, 11, 16315.2
Other major and minor comments are listed below:
- The title has to be rearranged highlighting the geographical areas and the comparison among different populations origin. Also, it would be better to add the complete species authority
- in the abstract, there are different mentions to habitat characteristics. It is my opinion that, without specific analyses on the relations between biological and environmental data, it is no possible to depict and indicate the effect of these latter on population features. Please try to re-arrange the abstract and the whole manuscript discussing results obtained from the analyses conducted
- In the keywords avoid the use of terms already used in the title
- In the introduction a research question is absent
- materials and methods section is very confuse, especially regarding the estuaries and sampling descriptions:
1: rearrange the section as follow; 2.1. Eels sampling; 2.2. Sampling areas description: 2.3. "Samples" processing etc...
2: provide a better map, including a wider area in an insert, and single estuaries pictures, as 3 of 6 estuaries are not visible in the current map
3: add the sampling permission and animal care strategies in this section
4: in presenting the reasons for using only right sagittae please consider to add data regarding the asymmetry performing new analyses using R-based shape analyses. It will also can be used for highlight differences among eels otolith from different estuaries
5: please add information on all the chemicals, instruments, and software used (product, brand, city, country)
6: please consider to present representative figures regarding otoliths analyses (radius, age, and growth)
- Results: please consider to add results of otolith shape analyses
- Discussion: please rearrange the discussion considering new data obtained from the shape analyses and avoid to discuss the possible roles of environmental parameters on differences among populations, or better, provide results and/or data on temperature, salinity, pH, and others, compared with the obtained results of this study
All the best regards
Round 2
Reviewer 1 Report
I thank the authors for considering the questions/comments/suggestions raised during the first review process. I am overall satisfied with it. I must congratulate the authors for their efforts done in reviewing the manuscript. Their adaptations and explanations seem sufficient to allow definitive acceptance for publication. As small suggestions there are:
L68 "In order to conserve and..." Not clear, please improve.
L233 Please adapt the y-axis length of CPUE to the standard deviation length of the Wimereux estuary for the year 2019 in Fig. 2(a)
Reviewer 2 Report
Dear Authors,
despite thanks for considering my comments, the manuscript has been very well re-arranged especially considering the addition of PCA analyses that was very important to highlight how different environmental parameters influence growth and eels populations structure.
All the best
Author Response
Dear Reviewer #2,
We would like to thank you for your valuable comments on our revised manuscript.
Best regards
Round 3
